# Differences in Maternal Immunoglobulins within Mother’s Own Breast Milk and Donor Breast Milk and across Digestion in Preterm Infants

**DOI:** 10.3390/nu11040920

**Published:** 2019-04-24

**Authors:** Veronique Demers-Mathieu, Robert K. Huston, Andi M. Markell, Elizabeth A. McCulley, Rachel L. Martin, Melinda Spooner, David C. Dallas

**Affiliations:** 1Nutrition Program, School of Biological and Population Health Sciences, College of Public Health and Human Sciences, Oregon State University, Corvallis, OR 97331, USA; Veronique.Demers-Mathieu@oregonstate.edu (V.D.-M.); spoonerm@oregonstate.edu (M.S.); 2Department of Pediatrics, Randall Children’s Hospital at Legacy Emanuel, Portland, OR 97227, USA; Robert_Huston@mednax.com (R.K.H.); amarkell@lhs.org (A.M.M.); emcculle@lhs.org (E.A.M.); rlmartin@lhs.org (R.L.M.)

**Keywords:** passive immunization, antibodies, lactation, prematurity, proteolysis, breast milk

## Abstract

Maternal antibody transfer to the newborn provides essential support for the infant’s naïve immune system. Preterm infants normally receive maternal antibodies through mother’s own breast milk (MBM) or, when mothers are unable to provide all the milk required, donor breast milk (DBM). DBM is pasteurized and exposed to several freeze–thaw cycles, which could reduce intact antibody concentration and the antibody’s resistance to digestion within the infant. Whether concentrations of antibodies in MBM and DBM differ and whether their survival across digestion in preterm infants differs remains unknown. Feed (MBM or DBM), gastric contents (MBM or DBM at 1-h post-ingestion) and stool samples (collected after a mix of MBM and DBM feeding) were collected from 20 preterm (26–36 weeks gestational age) mother–infant pairs at 8–9 and 21–22 days of postnatal age. Samples were analyzed via ELISA for the concentration of secretory IgA (SIgA), total IgA (SIgA/IgA), total IgM (SIgM/IgM) and IgG. Total IgA, SIgA, total IgM and IgG concentrations were 55.0%, 71.6%, 98.4% and 41.1% higher in MBM than in DBM, and were 49.8%, 32.7%, 73.9% and 39.7% higher in gastric contents when infants were fed with MBM than when infants were fed DBM, respectively. All maternal antibody isotypes present in breast milk were detected in the infant stools, of which IgA (not sIgA) was the most abundant.

## 1. Introduction

Mother’s own breast milk (MBM) provides maternal exposure-specific antibodies that provide passive immune protection to the infant. These maternal milk antibodies include IgA, IgG and IgM isotypes, as well as the secretory forms of IgA and IgM [1]. The antibodies in MBM that are ingested by preterm infants are comprised of ~80% total IgA (~73% SIgA/27% IgA), ~15% total IgM and ~5% IgG [1]. The provision of maternal milk antibodies helps compensate for the infant’s naïve immune system. Neonates have an immature intestinal immune system, as demonstrated by lower numbers of plasma cells (immune cells that can produce IgA, IgM and IgG) in colonic and rectal biopsies from term infants at 1–12 days postnatal age compared with those at 1 to 6 months postnatal age [2]. No study has demonstrated the specific time when the preterm infants are able to produce their own SIgA in the small intestine. Milk SIgA, which bind to and neutralize pathogens to prevent their adherence to epithelial cells and infection [3,4,5,6], provide important immune compensation.

Milk IgM and IgG may also play a role in infant intestinal mucosal defense. IgG can bind to viruses and prevent their attachment to the mucosal surface or trap pathogens when IgG binds to mucus [7]. However, IgG was less efficient than SIgA for altering attachment and trapping pathogens in mucin [7]. Though IgM-secreting cells have been identified in the infant gut, the role of IgM in infant mucosal immune defense remains unknown [2].

Preterm infants likely have an even less developed immune system than term infants. For example, from 1–28 weeks postnatal, preterm neonates (24–28 weeks of gestation) produce less diverse IgG antibodies (based on nucleotide sequences present in the variable region of the antibody gene as detected by RT-PCR) in their blood compared with term infants (36–42 weeks of gestation) [8]. As they are born early, preterm infants also miss some of the placenta–fetal IgG transfer that occurs for term infants [9]. For preterm infants, maternal milk antibodies may help compensate for the potentially lower secretion of antibodies, their loss of placenta–fetal IgG transfer time and perhaps lower immune function compared with term infants.

To neutralize pathogens in the preterm infant gut, maternal milk antibodies must survive digestive protease actions through the gastrointestinal tract to their site of action. Our recent studies demonstrated that milk total IgA concentration decreased by 60% from milk to the preterm infant stomach at 2-h post-ingestion, whereas total IgM and IgG were stable [1]. Two oral supplementation studies (in adults fed bovine colostrum SIgA/IgA, IgM and IgG [10] and in preterm infants fed serum IgA and IgG [11]) demonstrated that IgG and IgM survived intact to the stool, whereas SIgA/IgA did not. On the other hand, some studies have demonstrated that SIgA from MBM can survive to the infant stool and urine [12,13,14]. These oral supplementation studies did not determine the percentage of survival for SIgA, total IgA, total IgM and IgG to the infant stool. Moreover, measuring Igs in infant stool samples does not accurately represent the biological survival of Igs within the upper GI tract as they can be further degraded by colonic bacteria.

Preterm-delivering mothers often have difficulty making enough milk to feed their infants and often supplement with donor breast milk (DBM). Most mothers (72%) of very preterm infants (<27 week of gestational age (GA) are unable to provide all the MBM required, thus DBM is used to complete their diet [15]. Whether MBM and DBM differ in milk antibody concentrations remains unclear. DBM processing includes pooling milks from different mothers, Holder pasteurization (62.5 °C for 30 min) to inactivate viruses [16,17] and kill bacteria [18], and several freeze–thaw cycles. These factors could result in lower concentrations of maternal milk antibodies.

As the degree to which MBM and DBM antibody concentrations differ and how this affects the survival of antibodies during gastric digestion remains unknown, we examined these questions herein. Evidence of lower antibody concentrations in DBM could determine whether modification of the product or processing techniques are needed to improve infant health outcomes.

## 2. Materials and Methods

### 2.1. Participants and Sample Collection

#### 2.1.1. Participants and Enrollment

This study was approved by the Institutional Review Boards of Legacy Health Systems and Oregon State University. Samples were collected from twenty premature-delivering mother–infant pairs ranging in GA at birth from 26 to 36 weeks (Table 1) in the NICU. Eligibility criteria included having an indwelling naso/orogastric feeding tube, bolus feeding (<60 min infusion tolerated), feeding volumes of at least 4 mL and mothers who could produce a volume of MBM adequate for one full-volume feed per day. Exclusion criteria included neonates with diagnoses that are incompatible with life, gastrointestinal system anomalies, major gastrointestinal surgery, severe genitourinary anomalies and significant metabolic or endocrine diseases.

#### 2.1.2. Feeding and Sampling

In order to compare the concentration of immunoglobulins in DBM and MBM during preterm infant digestion, we gave two separate feedings of DBM and MBM without fortification rather than the typical feed consisting of a mixture of DBM and MBM with fortifier on days in which gastric sampling was accomplished. Milk and gastric samples (1–2 mL) were collected on 8–9 and 21–22 days of life. A separate sample of the donor milk (even though it was from 2 original pools) used to feed each infant was collected and used as biological replicates for all the comparisons between MBM and DBM. At both sample time periods, each infant received 2 of the normal 8 daily feedings as unfortified MBM or DBM on alternate days (randomized order). We randomized the order of feeding MBM and DBM to control for any potential effect of infant day of life on antibody digestion. The pool of DBM was acquired from two batches at Northwest Mother’s Milk Bank. Three-liter batches were pasteurized and frozen in 50-mL doses so that only a small fraction was thawed for each infant feeding. The power analysis based on detection of differences in antibody concentrations between MBM and gastric samples from preterm infants in our previous study [1] indicated that at least 15 infants were required to compare DBM and MBM-fed infants, as, in the previous study, most antibody concentrations differed significantly between milk and stomach with this sample size.

Prior to feeding, any gastric residuals were removed by syringe via the feeding tube to remove contamination from the previous feeding. Feedings were prepared at the Randall Children’s Hospital at Legacy Emanuel NICU using aseptic technique. Frozen MBM and DBM were thawed in Ameda Penguin warmers at 37 °C. Milk (either MBM or DBM) was fed to the infant via the nasogastric tube with a feeding pump set to deliver the entire bolus over 30–60 min. A 2-mL sample of the gastric fluid was collected 30 min after the completion of feed infusion. This sample collection timing was selected to match the gastric half-emptying time of premature infants to maximize time in the stomach as well as our ability to collect remaining contents [19]. As the mouth and esophagus do not contribute to proteolytic digestion, the use of nasogastric tubes (bypassing this) will likely not alter the results from an enteral feed taking orally. After collecting the 2-mL gastric sample, 2 mL of additional feed plus the additional volume recorded of gastric residue that was removed prior to the feed were provided to avoid any nutritional interruption. Stool (1 g) was collected within 48 h of the gastric sampling time point and was recovered from the diaper and scraped into a sterile jar. Stool sample collection was not specific to DBM/MBM and thus represents stools deriving from a mixture of DBM and MBM feeding. After collection of each sample type (feed, gastric and stool), samples were placed immediately on ice and stored at −80 °C in the NICU. Samples were then be transported on dry ice to Oregon State University for sample analysis.

#### 2.1.3. Clinical Data Collection

Infant GA and postnatal age at mother’s milk feeding and at donor breast milk feeding were collected (see Table 1).

### 2.2. Sample Preparation and ELISAs

Feed (MBM and DBM) and gastric samples were thawed at 4 °C, pH was determined and samples (1 mL) were centrifuged at 4000× *g* for 20 min at 4 °C. The infranate was collected, separated into aliquots (100 μL) and stored at −80 °C. Frozen stool samples (0.1 g) were diluted in 700 μL of phosphate-buffered saline pH 7.4 (Thermo Fisher Scientific, Waltham, MA, USA) with 0.05% Tween-20 (Bio-Rad Laboratories, Irvine, CA, USA) (PBST) and 3% fraction V bovine serum albumin solution (Innovative Research, Novi, MI, USA). Diluted stool samples were mixed by vortex for 2 min and then the vials were centrifuged at 4000× *g* for 20 min at 4 °C. The supernatant was collected, separated into aliquots (100 μL) and stored at −80 °C. Sample pH measurements were performed with an S220 SevenCompact pH/Ion meter (Mettler-Toledo) equipped with a combined sealed glass electrode.

The spectrophotometric ELISAs were measured with a microplate reader (Spectramax M2, Molecular Devices, Sunnyvale, CA, USA) with two replicates of blanks, standards and samples. SoftMax Pro 7.0 Microplate Data Analysis Software (Molecular Devices) was used to create a standard curve with a Four-Parameter Logistic curve fit. Clear flat-bottom Immuno 96-well plates MaxiSorp (Thermo Fisher Scientific) were coated with 100 μL of 1 μg/mL of capture antibodies: goat anti-human IgA alpha-chain for SIgA or total IgA (SIgA/IgA); rat anti-human IgM mu-chain for total IgM (SIgM/IgM) and goat anti-human IgG gamma-chain for IgG (Bio-Rad Laboratories). Plates were incubated overnight at 4 °C. After incubation, plates were washed 3 times with PBST and then 100 μL of blocking buffer (PBST with 3% of fraction V bovine serum albumin solution) was added in all wells for 1 h at room temperature. Standard samples were prepared using purified IgA from human colostrum (Sigma-Aldrich, St. Louis, MO, USA) for SIgA and total IgA, and purified IgM and IgG from human serum (Sigma-Aldrich). The standard curves were prepared using a dilution series of standard antibody in blocking buffer and the final concentration covered a range from 1 to 5000 ng/mL. Feed (MBM or DBM) and gastric samples were diluted 250× with blocking buffer for total IgM and IgG measurements and 500× for SIgA and total IgA measurements. Prediluted stool samples were diluted in 250× for total IgA, total IgM and IgG, and 20× for SIgA. For each step (addition of standards/samples and secondary antibodies at 1 μg/mL), washing and incubation for 1 h at room temperature were performed. To determine SIgA concentration, mouse anti-human IgA secretory-chain was added, the plate was washed and anti-mouse IgG:horseradish peroxidase (HRP) was added. To determine total IgA concentration, goat anti-human IgA alpha-chain:HRP was used. For total IgM, goat anti-human IgM mu-chain:HRP was used. For IgG, goat anti-human IgG gamma-chain: HRP was used (all antibodies from Bio-Rad). The substrate (1×, 100 μL), 3,3′,5,5′-tetramethylbenzidine (Thermo Fisher Scientific), was added for 5 min at room temperature followed by addition of 50 μL of 2 N sulfuric acid to stop the coloration reaction. Optical density was measured at 450 nm.

The protein concentration of all samples was measured with the BCA protein assay (Thermo Fisher Scientific). Feed (MBM or DBM) and gastric samples were diluted 10× in the diluent provided in the kit, whereas dissolved stool samples were diluted 250×.

### 2.3. Pasteurization and Freezing/Thawing Effects on Mother’s Milk Antibodies

To directly test the effect of pasteurization and freeze–thaw cycles on mother’s milk antibodies, 500 mL of milk from one mother who delivered a term infant at 38 weeks of gestational age was pumped and collected in a sterile plastic bag at 12 days of postnatal age. Six 1-mL aliquots were centrifuged at 4000× *g* for 30 min at 4 °C and the infranates were collected (skim milk). Three of these skim milk samples were used for the control raw human milk (RHM) and 3 others were pasteurized at 62.5 °C for 30 min for pasteurized skimmed human milk (PSHM). Three other 1-mL aliquots of whole human milk were pasteurized without skimming (pasteurized whole human milk, PWHM). After pasteurization, the PWHM samples were centrifuged at 4000× *g* for 30 min at 4 °C to remove the lipid layer. Each sample was analyzed via ELISAs for SIgA, total IgA, total IgM and IgG.

To test the effect of freeze–thaw cycles, three 1-mL aliquots of whole raw milk were frozen at –20 °C, 3 aliquots were frozen at –80 °C and 3 aliquots were stored on ice for 1 h. After 1 h in the freezer, the samples were thawed rapidly at 37 °C. All samples were centrifuged at 4000× *g* for 30 min at 4 °C to remove the lipid layer and analyzed via ELISAs for SIgA, total IgA, total IgM and IgG.

### 2.4. Statistical Analyses

Student’s *t*-tests were used to determine whether the measurements in samples differed between 8–9 and 21–22 days of postnatal age. As the concentrations of total IgA, SIgA, total IgM, IgG and pH did not differ between 8–9 and 21–22 days of postnatal age (see Appendix A for statistical analyses), these samples were combined to compare groups by type of feeding (MBM versus DBM) in milk and gastric samples within the same mother–infant pairs using Wilcoxon matched-pairs signed-rank test in GraphPad Prism software (version 7.03). All tests were nonparametric as some of the values did not pass the D’Agostino and Pearson normality test. Wilcoxon matched-pairs signed-rank test was also used to compare total protein concentration in samples at 8–9 and 21–22 days of postnatal age. Though Student’s *t*-tests did not reveal differences between days 8–9 and 21–22 days of postnatal age, we performed Wilcoxon matched-pairs signed-rank test to compare antibody concentrations in MBM and DBM in milk and gastric samples at 8–9 days and 21–22 days separately, which are shown in the Appendix A. Though our study was not designed to have statistical power to compare infants by GA groups, we examined whether antibody concentrations in the samples differed by GA groups (26–27 week, 30–31 week and 35–36 week) using Wilcoxon matched-pairs signed-rank test, which is shown in the Appendix A. We also evaluated the effect of the antibiotics that infants received or not in gastric and stool samples from both feedings (MBM and DBM) using Student’s *t*-tests, which is shown in the Appendix A. One-way ANOVA followed by Dunnett’s multiple comparisons test were performed to compare RHM with PWHM and PSHM as well as raw milk to frozen milks. Differences were designated significant at *p* < 0.05.

## 3. Results

### 3.1. Infant Demographics

The demographic details for the preterm-delivering mother–infant pairs are presented in Table 1.

### 3.2. pH

The pH values of MBM (average 7.27 ± 0.05) were significantly higher than of DBM (average 6.57 ± 0.02, *p* < 0.001, Figure 1A), but they did not differ in gastric contents when infants were fed MBM or DBM (average 5.2 ± 0.1, Figure 1A). Both MBM and DBM had higher pH values than their respective gastric samples (*p* < 0.001) (Figure 1A).

### 3.3. Protein Concentration

Protein concentration in MBM collected at 8–9 days of life (average 20.2 ± 0.9 mg/mL) was 1.3-fold higher than that in MBM collected at 21–22 days of life (average 15.6 ± 0.9 mg/mL, *p* < 0.001). Protein concentration in the stomach (average 11.2 ± 0.6 mg/mL) did not differ between 8–9 and 21–22 days for MBM or DBM. Protein concentration in MBM was 1.9- and 1.6-fold higher than in DBM at 8–9 (average 10.2 ± 0.4 mg/mL) and 21–22 days (average 9.4 ± 0.6), respectively (*p* < 0.001, Figure 1B). Protein concentration in the stomach from preterm infants fed MBM (average 13.2 ± 0.7 mg/mL) was 1.6-fold higher than those fed DBM (average 9.4 ± 0.8 mg/mL, *p* < 0.01, Figure 1B). Protein concentration decreased 1.3-fold from MBM to gastric contents from infants fed MBM at 8–9 days (*p* < 0.01, Figure 1B) but did not differ at 21–22 days of postnatal age (Figure 1B). Protein concentration decreased 1.2-fold from DBM to gastric contents from infants fed DBM at 8–9 and 21–22 days postnatal age (combined across days, *p* = 0.007, Figure 1B).

### 3.4. Maternal Milk Antibody Concentrations

#### 3.4.1. Antibody Concentrations in MBM and DBM

Total IgA, SIgA, total IgM and IgG concentrations were respectively 55.0, 71.6, 98.4 and 41.1% higher in MBM compared with DBM (*p* < 0.001, Figure 2A–D).

#### 3.4.2. Maternal Antibody Digestion

Total IgA, SIgA, total IgM and IgG concentrations were respectively 49.8, 32.7, 73.9% and 39.7% higher in gastric contents from infants fed MBM than infants fed DBM (Figure 2A–D).

SIgA and total IgM concentration significantly decreased (Figure 2B,C) 28.8 and 39.8%, respectively, from MBM to the preterm infant stomach but did not change for total IgA and IgG (*p* > 0.05, Figure 2A,D). Total IgA, SIgA, total IgM and IgG concentrations did not change from DBM to the preterm stomach (Figure 2A–D).

In MBM, the proportions of total IgA, total IgM and IgG were respectively 79.4, 18.1 and 2.4%. In the gastric contents after feeding with MBM, the proportions of total IgA, total IgM and IgG were respectively 84.4, 12.2 and 3.3% (Figure 3A). The proportion of total IgA that was SIgA decreased from 80% in MBM to 60% in the gastric samples, whereas this proportion increased from 51% in DBM to 81% in the gastric samples.

Total IgA, SIgA, total IgM and IgG were detected in stools from preterm infants (Figure 3). These antibodies could derive from the MBM and/or DBM feedings. Antibody concentration did not differ in stool between 8–9 and 21–22 days of postnatal age. The proportion of total IgA, total IgM and IgG was respectively 85.2, 12.7 and 2.1%, whereas SIgA was only 0.7% of total IgA (Figure 3). The proportion of SIgA in stool was much lower than that present in milk and the gastric contents.

#### 3.4.3. Pasteurization and Freeze–Thaw Effects

The concentration of total IgM decreased 62% from RHM to PWHM but those of SIgA, total IgA and IgG did not differ (Appendix A). Concentrations of total IgA, SIgA, total IgM and IgG decreased 37, 23, 87 and 54%, respectively from RHM to PSHM.

For all isotypes, the concentration did not differ between fresh milk and that exposed to freeze–thaw cycles at −20 or −80 °C (*p* > 0.05).

## 4. Discussion

The present study examined whether the concentration of maternal milk antibodies differed between MBM and DBM and across their digestion in the stomach of preterm infants. We also measured the survival of milk antibodies to the infant stool. For the first time, we demonstrated that the concentrations of all the Ig isotypes (total IgA, SIgA, total IgM and IgG) were higher in MBM than in DBM. The apparent lower antibody concentrations in DBM could be due to milk processing (including Holder pasteurization), the different gestational age at which mothers delivered and different postnatal day at which milks were expressed. Total IgM concentration was strongly reduced in both pasteurized whole milk and skim milk, whereas total IgA, SIgA and IgG concentrations were reduced in pasteurized skim milk but not in pasteurized whole milk. Our results that IgM was more sensitive to Holder pasteurization than SIgA agree with those of Ford et al. [20] who reported that SIgA from centrifuged MBM (pooled mature milk from term-delivering mothers) decreased 22% after Holder pasteurization, whereas IgM concentration had 100% loss (IgG was not measured). This observation was in accordance with our results in that the difference between MBM and DBM was the highest for IgM.

A previous study demonstrated that IgA and IgG concentrations were stable after freezing MBM at −20 °C for 3 months [21]. In contrast, Pardou et al. [22] showed that total IgA concentration in MBM from nursing mothers in the maternity ward and neonatal unit decreased 17.6% after freezing MBM at −20 °C for 4 days followed by thawing when contaminating bacteria were detected in milk via culture methods but not when those bacteria were absent. IgA proteases produced by some microbes (e.g., viridans streptococci and *Bacteroides* spp.) can cleave peptide bonds in the hinge region of the IgA1 heavy chain and then reduce their concentration in contaminated milk [22]. Our investigation showed that IgA, SIgA, IgM or IgG concentrations were maintained after freeze–thaw cycles at −20 and −80 °C in human milk. Therefore, the effect of freeze–thaw cycles is likely minimal if the milk is not contaminated but could reduce IgA1 concentration if the milk is contaminated. Before pasteurization, MBM is thawed at least twice and it is manipulated by the donors (pumping milk) when making DBM, which could lead to contamination with bacteria that produce these extracellular peptidases that could degrade proteins.

As a baseline validation, protein concentration was lower in DBM than in preterm milk, which validates the results of a previous study [23]. DBM is often donated from term-delivering mothers at late lactation time, whereas the MBM examined in this study was from preterm mothers at early lactation time, which may have higher antibody concentrations. We observed that the concentration of total IgA or total secretory component (free secretory component (SC)/SIgA/SIgM) did not differ between preterm milk and term milk from 6 to 28 days of postnatal age [1], and this observation was also demonstrated by another study [24]. However, Chandra et al. [25] found that IgA concentration in preterm milk was higher than in term milk from 3 to 15 days of postnatal age. Total IgA in colostrum (1–3 days of postnatal age) from preterm-delivering mothers was higher than in colostrum from term-delivering mothers in several studies [24,25,26]. Total IgA concentration in preterm milk decreased from 1 week to 2 months of postnatal age [1] and from colostrum to mature milk [24] but did not differ across time in term milk [1]. IgM and IgG did not differ across time postpartum in preterm and term milk [1]. IgM and IgG concentrations in preterm-delivering mothers were lower than [1], higher than [25] or similar to [27,28] that in term-delivering mothers. Therefore, the observed higher total IgA concentration in MBM compared with DBM could be due to the early lactation time of the preterm-delivering mothers. We observed a decrease in total IgM concentration with increasing GA, but no correlation between GA and total IgA, SIgA or IgG concentrations. The present study found a high variation (SD) in antibody concentration for all isotypes among mothers (possibly due to maternal background factors, such as vaccination schedule, diets, etc.). These variations could be responsible for the differing results between studies.

The proportion of SIgA and IgA (without SC) in milk from premature-delivering mothers as well as the slightly higher proportion of IgG compared with IgM are in agreement with our previous study where the proportion of antibody in preterm-delivering mothers was 82% for total IgA, 60% for total secretory component (SC/SIgA/SIgM), 6.9% for total IgM and 11.0% for IgG [1]. Percentages of SIgA in total IgA also matched the observations reported by Goldman et al. [29] that total SC (called “SIgA” by the authors but actually representing total SC, as an anti-SC primary antibody was used) concentration represented 90% of the total IgA in milk from term-delivering mothers. We also found a similar proportion of these isotypes in the gastric samples from infants fed MBM compared with the MBM samples (feeds).

For the first time, this investigation reported that total IgA, SIgA, total IgM and IgG concentrations were higher in the stomach from preterm infants fed MBM than those fed DBM. This higher concentration could be due to the initial higher concentration of maternal antibodies in MBM compared with DBM. SIgA and total IgM in MBM were partially digested in the stomach, total IgA and IgG in MBM were stable in gastric contents and none of the isotypes in DBM were digested. The lower apparent digestibility of antibodies in DBM could be due to the changes in the Ig structure after pasteurization. A previous study observed that pasteurization of breast milk enhanced the gastric digestion of lactoferrin and reduced that of alpha-lactalbumin [30]. Protein susceptibility to protease (pepsin or milk proteases) is influenced by the specific protein structure in the milk emulsion [31]. Pasteurization could change the organization of Igs and reduce the accessibility of cleavage sites to proteases (i.e., milk proteases or pepsin).

Gastric digestion reduced the amount of SIgA and total IgM from MBM but total IgA and IgG from MBM were not affected. No milk antibody isotype decreased significantly from DBM to the gastric contents. SIgA, IgM and IgG after DBM feeding appeared to be slightly, but not significantly, higher in the gastric samples. This observation could be due to contamination of the gastric contents with residual from previous feeds based on MBM.

A small degradation of SIgA from MBM in the preterm infant stomach was observed, which is similar to findings of our previous study where the decrease of total IgA was likely derived from preterm gastric digestion of partly IgA and partly SIgA [1]. IgM was digested but total IgA and IgG were stable in the present study, which differed from our previous findings that preterm infants partially degraded total IgA but not IgG and IgM in the stomach [32]. These differences could be due to the longer postprandial time (2-h post feed initiation) in the previous study [32] compared with the present study (30 min post feed completion). The reduction of SIgA and total IgM in gastric contents is due to the degradation by proteases (pepsin and/or milk proteases [33]) and not acid-induced denaturation in the stomach, as we previously demonstrated that standard IgA and IgM did not decrease in concentration in gastric acid conditions [1].

Antibodies detected in stool samples could derive from the MBM and/or DBM feeding or be potentially generated by the infant. The proportions of Igs from MBM and DBM in infant stool differed significantly from those in feed and gastric contents, especially the proportion of total IgA made up by SIgA. Unlike MBM and DBM samples in which most of the total IgA was SIgA, most of the total IgA in stool was IgA without the SC. IgM and IgG were detected in the stools at concentrations lower than total IgA. The lower proportion of SIgA compared to total IgA in the stool was likely due to release of the SC portion during gastrointestinal proteolytic digestion. A few studies have measured the survival of milk Igs to infant stools [11,34]. A greater reduction of IgA compared with IgG or IgM in preterm infant stools was previously observed [11]. These investigators fed preterm infants (1–28 days of postnatal age (GA unknown) 0.8–2 kg BW) infant formula plus pasteurized pooled breast milk supplemented with 600 mg daily of serum-derived human IgA (non-secretory form) (73%) and IgG (26%). The stool samples collected contained 1–10 mg IgG per g of dried feces (percentage reduction not calculated) and no IgA [11].

Five infants among the total of twenty received antibiotics (ampicillin/cefdinir) (Table 1). Provision of antibiotics could change the microbiome and alter the survival of milk antibodies to the stool. However, no differences were detected in the survival of Ig in the stomach and stool between infants with and without antibiotics (Appendix A). We did observe that SIgA concentration tended to be higher in the gastric contents (after MBM feeding) from infants that did not receive antibiotics compared with those that received antibiotics.

The gastric pH was similar between infants fed MBM and those fed DBM even though the initial pH of MBM was higher than that of DBM. A previous study demonstrated that pasteurization of breast milk did not affect its pH [30], whereas storage at −20 °C and thawing reduced its pH [35], likely because of lipolytic release of free fatty acids [36]. The lower pH in DBM compared with MBM is likely due to repeated freeze–thaw cycles during milk processing.

One limitation of this study is the collection of stool samples within a 48-h window after collecting the milk and gastric samples. This window was selected for convenience to collect a stool sample around the time of milk and gastric sample collections. We could not determine whether stool samples derived from the MBM or DBM feeding, thus they represent a mixture of the total diet. Future studies could attempt to separate DBM and MBM-derived stools using indigestible markers, such as food colorants. Moreover, the immunoglobulins detected in the stool could also derive from the infant’s own production. Though IgA-, IgG- and IgM-positive plasma cells are low in the lamina propria of the rectum and colon of term infants in the first month, they are indeed present and could contribute to the appearance of antibodies in the stool. In order to confirm that immunoglobulins in stool derive from feed (MBM or DBM) rather than the infant would require a protein labeling approach. The use of a single human milk sample to examine the effect of pasteurization and freeze–thaw cycles on the retention of antibodies could also represent a limitation. We used a single sample in order to focus on the effect of treatment; however, variations in biological samples could modify the treatment effects. Another limitation is that we did not have enough subjects to detect many differences between MBM and DBM feeds and gastric contents when grouped by GA (26–27 weeks, 30–31 weeks and 35–36 weeks) (Appendix A). Moreover, we did not have enough subjects to detect differences between days 8–9 and 21–22, though a larger sample size could have identified differences as many p-values for MBM were close to being significantly different (Appendix A).

These findings indicate that the concentration of maternal antibodies in DBM may be lower than in MBM. If that is the case, preterm infants fed DBM may receive lower amounts of antibodies than infants fed MBM. Lower ingestion of milk antibodies could result in lower protection against infections in these vulnerable infants. Indeed, several studies may provide some evidence of lowered protection. A clinical study of 226 infants observed that preterm infants fed raw MBM tended to have lower infection rates than those fed pasteurized MBM (10.5% versus 14.3%) [37]. Another study of 243 preterm infants showed that infants fed MBM had fewer episodes of late-onset sepsis, necrotizing enterocolitis, total infection-related events, shorter hospital stay duration and had a lower percentage of blood samples that tested positive for Gram-negative bacteria than infants fed DBM [38].

## 5. Conclusions

The present study revealed that antibody concentrations (total IgA, SIgA, total IgM and IgG) were higher in MBM from preterm-delivering mothers than in DBM. Their concentrations in gastric contents were higher from infants fed MBM than those fed DBM, but Igs from DBM were less digested than Igs from MBM. All isotypes were detected in stools from preterm infants, but the proportion of Ig made up by SIgA decreased dramatically; most surviving Ig were IgA (without SC). The higher concentration of antibodies in MBM than in DBM may make MBM more effective in preventing enteric pathogens adhesion and invasion in newborns compared with DBM. This information could lead to changes in processing and collection practices to preserve these immune components in DBM.

## Figures and Tables

**Figure 1 nutrients-11-00920-f001:**
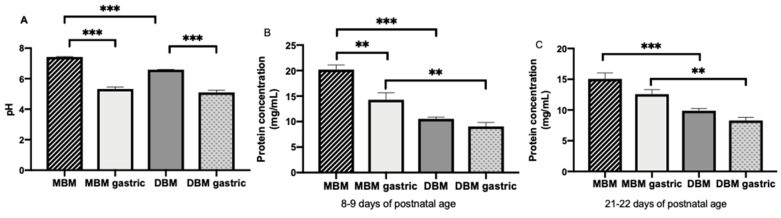
(**A**) pH values of milk and gastric contents at 1-h postprandial time from 20 preterm infants (26–36 weeks of gestational age (GA)) fed mother’s own breast milk (MBM) and donor breast milk (DBM). Values are mean ± SEM, *n* = 36 for each group (*n* = 20 for 8–9 days and *n* = 16 for 21–22 days of postnatal age). (**B**) Total protein concentration in milk and gastric contents from infant fed MBM and DBM at 8–9 days (*n* = 20) (**C**) Total protein concentration in milk and gastric contents from infant fed MBM and DBM at 21–22 days (*n* = 16). Asterisks show statistically significant differences between variables (*** *p* < 0.001; ** *p* < 0.01) using the Wilcoxon matched-pairs signed-rank test.

**Figure 2 nutrients-11-00920-f002:**
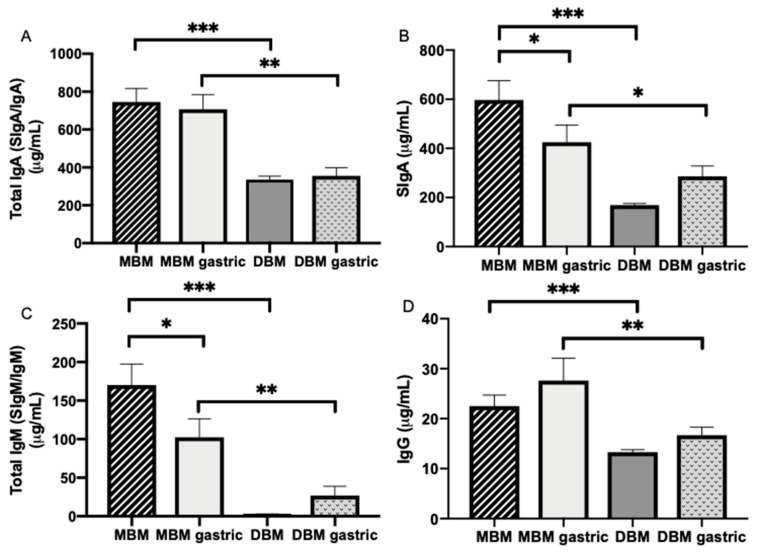
Immunoglobulin concentrations in milk and gastric contents at 1-h postprandial time from 20 preterm infants (26–36 weeks of gestational age (GA)) fed mother’s own breast milk (MBM) and donor breast milk (DBM). Concentration of (**A**) total IgA (SIgA/IgA), (**B**) secretory IgA (SIgA), (**C**) total IgM (SIgM/IgM) and (**D**) IgG in milk and gastric samples. Values are mean ± SEM, *n* = 36 for MBM and DBM (*n* = 20 for 8–9 days and *n* = 16 for 21–22 days of postnatal age). Asterisks show statistically significant differences between variables (*** *p* < 0.001; ** *p* < 0.01; * *p* < 0.05) using the Wilcoxon matched-pairs signed-rank test.

**Figure 3 nutrients-11-00920-f003:**
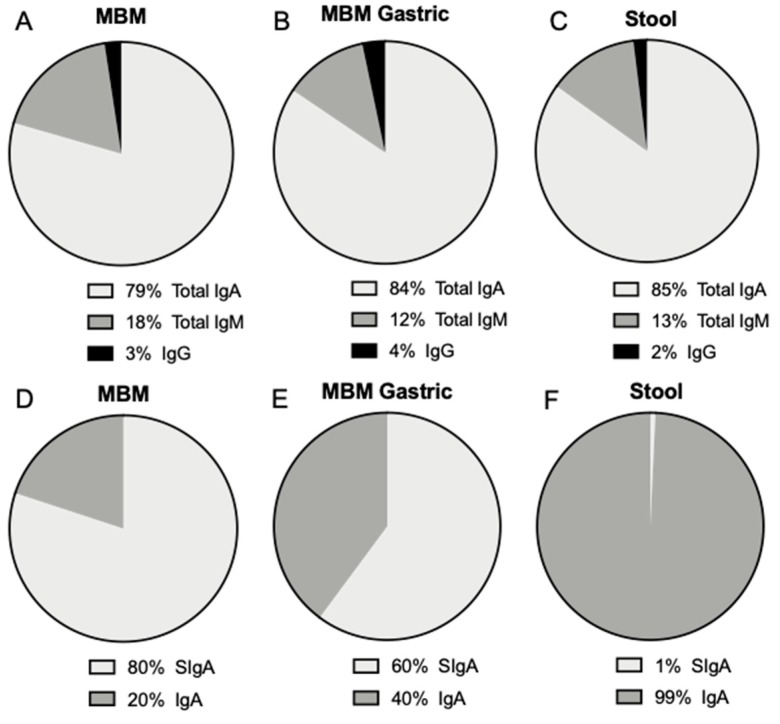
Proportion of total IgA, total IgM and IgG (**A**) in mother’s own breast milk (MBM), (**B**) in gastric contents at 1-h postprandial time from preterm infants fed MBM and (**C**) in infant stools (from MBM, DBM and/or infant self). Proportion of SIgA and IgA from total IgA (**D**) in mother’s own breast milk (MBM), (E) in gastric contents at 1-h postprandial time from preterm infants fed MBM and (**F**) in infant stools. Values are mean, *n* = 36 for MBM (*n* = 20 for 8–9 days and *n* = 16 for 21–22 days of postnatal age).

**Table 1 nutrients-11-00920-t001:** Demographics of preterm-delivering mother–infant pairs sampled for mother’s own breast milk, gastric contents (1-h postprandial time) and stools (24-h post-feeding).

Demographics	Preterm-Delivering Mother Infant Pairs ^1,2^
GA, week	30 ± 3 (26–36)
Postnatal age at mother’s milk feeding, days	8.6 ± 0.1 and 21.4 ± 0.1
Postnatal age at donor breast milk feeding, days	8.4 ± 0.1 and 21.6 ± 0.1
Birth weight at birth, kg	1.5 ± 0.7 (0.7–3.6)
Weight gain velocity, g/kg/day ^3^	11 ± 4
Length gain velocity, cm/week ^3^	1.0 ± 0.3
Head circumference gain velocity, cm/week ^3^	0.6 ± 0.3
Volume of feeding at 8–9 days of postnatal age, mL	16 ± 12 (2.5–38)
Volume of feeding at 21–22 days, of postnatal age, mL	28 ± 10 (14–49)
Infant sex	10 females; 10 males
Infant of a diabetic mother, *n*	3
SGA10, less than the 10th percentile ^4^, *n*	2
SGA3, less than the 3rd percentile ^4^, *n*	1
Intrauterine illicit drug exposure, *n*	2
C-section, *n*	16
Retinopathy of prematurity, *n*	0
Chronic lung disease, *n*	2
Gram-negative sepsis, *n*	1
Gram-positive sepsis or fungal sepsis	0
Late-onset sepsis	1
Antibiotics^5^ received postnatally, *n*	5
Necrotizing enterocolitis	0
Gastrointestinal bleeding	0
Cardiopulmonary resuscitation	0
Infant death	0
Length of stay in NICU	47 ± 29 (10–109)

^1^ Values are mean ± SD (range); ^2^ Number of paired milk and gastric and stool samples from 20 preterm infants was *n* = 20 at 8–9 days and *n* = 16 at 21–22 days of postnatal age; ^3^ Birth to discharge; ^4^ Small for gestational age (SGA) on the Fenton 2013 Growth chart; ^5^ Antibiotics were ampicillin/cefdinir.

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
