# Peer review of "Differences in Maternal Immunoglobulins within Mother’s Own Breast Milk and Donor Breast Milk and across Digestion in Preterm Infants"

_nutrients, 2019, doi:10.3390/nu11040920_

Round 1
Reviewer 1 Report
Differences in maternal Immunoglobulins within Mother’s Own Breast Milk and Donor Breast Milk and Across Digestion in Preterm Infants
This article is an important contribution to the literature as it is important to emphasize the benefits of mother’s own milk for preterm infants.
Major concerns:
1. No mention of limitations of the study. Need to add a paragraph about limitations. They assumed transit time to be ? 48 hours since they collected their stools at this time point. What if digestion or transit time was faster or slower? Authors need to mention this weakness and how they accounted for it.
2. Further description is needed as to why they randomized EBM or DEBM line 104
3. I strongly feel that they should not include “2.3 Pasteurization and freezing/thawing effect on mother’s milk antibodies” and “3.4.3 Pasteurization” as there was only 1 donor mom that they studied. Although the milk was aliquoted, there is too much bias using only 1 donor mom. I recommend either excluding this section from the paper (it is distracting from the main point) or they add more data and samples to the section (instead of one donor mom).
4. Authors need to explain sample size- why did they chose the current sample size?
5. Why are all figures combining the two time points? I strongly recommend the time points being separated as lactation could be very different at these time points ( 8-9 days vs. 21-22 d). Authors do not mention why time points combined and then no mention in limitations of this method.
Minor comments:
Line 76: Evidence of lower antibody concentration in DBM could suggest whether modification of the product or processing techniques are needed to improve infant health outcomes. I would use: determine if or possibly support
Line 99: d- I feel like days should be spelled out (throughout article)
Page 3- Does the table need a heading: Table 1? ( Line 194 mentions Table 1.)
Line 311: There are 2 periods after delivering mothers.
Line 362: A previous studies- should say study?
Author Response
Reviewer 1
Differences in maternal Immunoglobulins within Mother’s Own Breast Milk and Donor Breast Milk and Across Digestion in Preterm Infants
This article is an important contribution to the literature as it is important to emphasize the benefits of mother’s own milk for preterm infants.
Major concerns:
1. No mention of limitations of the study. Need to add a paragraph about limitations. They assumed transit time to be ? 48 hours since they collected their stools at this time point. What if digestion or transit time was faster or slower? Authors need to mention this weakness and how they accounted for it.
>> We did not assume a transit time of 48 h. We indicated that stool samples were collected within 48 h after the first feed and that its contents derive from a mixture of their different feeds. We added this statement to the Methods section (lines 151-153). We also added a sentence in the discussion to point out that antibodies detected in stool samples could derive from the MBM and/or DMB feeding or other previous feedings (line 389-390).We also added that a paragraph about the limitations of this study, in which we specified that we could not collect based on a specific feed in this study which is why we used a large 48-h convenience sampling window, and that future studies may be able to separate stools deriving from specific feeds using indigestible markers, such as food colorants (lines 413-418).
2. Further description is needed as to why they randomized EBM or DEBM line 104
>> Thank you. We added an explanation for the randomization that we used. We randomized the order of MBM and DBM to control for any potential effect of the infant day of life on antibody digestion (lines 131-133).
3. I strongly feel that they should not include “2.3 Pasteurization and freezing/thawing effect on mother’s milk antibodies” and “3.4.3 Pasteurization” as there was only 1 donor mom that they studied. Although the milk was aliquoted, there is too much bias using only 1 donor mom. I recommend either excluding this section from the paper (it is distracting from the main point) or they add more data and samples to the section (instead of one donor mom).
>> We wanted to include this point to help explore some of the differences observed between donor milk and mother’s milks. We actually selected to use just a single donor milk sample because we are studying the effect of the treatments, not of the specific milk. We do realize that your point is valid, in that changes due to treatment could have some variation based on the specific milk perhaps because of complex interactions. We’d like to keep these points in as we feel that it helps to explain what we found in the paper. We did change Figure 4 to Figure S3 in order to reduce potential distraction from the main point. Moreover, we have added this point as a limitation of the study design in the Discussion section (lines 418-421).
4. Authors need to explain sample size- why did they chose the current sample size?
>> We added that our power analysis based on detection of differences in antibody concentrations between MBM and gastric samples from preterm infants in our previous study demonstrated that each sample group should contain at least 15 infants, as most antibody concentrations differed significantly between milk and the stomach with this sample size (lines 135-138).
5. Why are all figures combining the two time points? I strongly recommend the time points being separated as lactation could be very different at these time points (8-9 days vs. 21-22 d). Authors do not mention why time points combined and then no mention in limitations of this method.
>> We explained in the statistical analysis section that concentrations of total IgA, SIgA, total IgM, IgG and pH did not differ between 8–9 and 21–22 d of postnatal age (Table S1) and that for that reason, these samples were combined to compare groups by type of feeding (MBM versus DBM) (see line 216-220). Table S1 shows that p-values of the t-tests that compared these day timepoints, showing that none except for protein concentration within MBM differed between day 8-9 and day 21-22. To address your comment, we added a supplemental Figure (Figure S1) showing the data with days 8-9 and 21-22 separated. In this figure, you can see that there are no statistically significant differences between days 8-9 and days 21-22. We also added that the relatively small sample size in this study may have limited our ability to detect significant differences between days 8-9 and 21-22, as several of the p-values for MBM were close to significance (lines 423-427).
Minor comments:
Line 76: Evidence of lower antibody concentration in DBM could suggest whether modification of the product or processing techniques are needed to improve infant health outcomes. I would use: determine if or possibly support
>> We added “determine” in the sentence (line 99).
Line 99: d- I feel like days should be spelled out (throughout article)
>>We spelled out “days” throughout the article.
Page 3- Does the table need a heading: Table S1? (Line 194 mentions Table S1.)
>> Table S1 has a heading and is in the Supplement data, which is different from Table 1 in the main article.
Line 311: There are 2 periods after delivering mothers.
>> Thank you.
Line 362: A previous studies- should say study?
>> We corrected this.
Reviewer 2 Report
Demers-Mathieu et al present a study that aimed to evaluate differences in immunoglobulin concentrations in human milk, gastric residuals and stool samples of preterm infants in relation to feeds of either mother’s own milk (MBM) or donor breast milk (DBM). The research objective is important since an impaired immune function is an important risk factor in prematurity.
Previously the same group presented comparable research, evaluating the effect of gastric digestion on either enzymatic activity, proteins and immunoglobulins in preterm and term infants. The current study adds the evaluation of donor breast milk and stool samples. Authors previously concluded that human milk total SC (SIgA/SC/SIgM), IgM, and Ig G, except IgA survived mostly intact after gastric digestion, especially in preterm infants. The current study found a lower amount of immunoglobulins in DBM compared to MBM, which was also seen in gastric samples. Somewhat in contrast to previous findings IgA was found most abundant in stools.
While the paper is well structured and easy to read, a number of major comments can be made with regard to theoretical assumptions and design of the study.
With regard to design: 20 and 16 paired samples from preterm infants with a gestational age ranging from 26 to 36 weeks were evaluated at 2 time points. This is a very small number taking into account the great maturational differences between very preterm infants and nearly term infants. Thus results may be biased by great interindividual differences. A power calculation is not presented.
Further, it is well- known that bio-active factors in human milk differ between mothers of very-/pre- and term infants and the content of these factors changes during the lactation period. A lower content of protein and immune-factors in DBM has also been demonstrated many years ago and is one of well-known disadvantages of DBM. Authors themselves previously demonstrated that gastric digestion will affect the feed content, thus in this respect the study does not present new insights.
Immunoglobulins in stool: while analysis of feed and gastric residuals are based on a single feeding event at two time points the stools are collected within 48 hrs after the feeding event. How do authors guarantee that the immunoglobulins measured in stool are related to the single feeding? How do authors know that the immunoglobulins found in stool originate from MBM of DBM and not form the infant self? While it is known that the amount of immunoglobulins in preterm infants usually is lower than in termborns or adults it has also been demonstrated that human milk contains a number of bio-active factors that probably stimulate the production of immunoglobulins and thereby may increase the amount of own infant immunoglobulins. This seems in contrast to author’s assumption in the introduction that intestinal mucosal plasma cells only begin producing SIgA after 1 mo of age. Further, this assumption is not confirmed by reference [3]. Probably, to demonstrate the survival of immune-factors into stool radioactive labelling would be needed.
Materials and Method
- range of prematurity: did author consider to evaluate sub groups base on Ga for instance 26-28, 29- 30, 30 – 32. However this would probably have meant to include a higher number of patients.
- eligibility: how many infants were suspected for EOS and received antibiotics postnatally? This may have changed the intestinal microbiome and the amount of intestinal bio-active factors.
Line 99 with fortifier on ‘d’? probably day
- how many ml feeding did infants receive?
Stool collection time: see above , motivation for this procedure is needed. Was the first stool after the event collected?
Statistical analyses
- as mentioned above it is very strange that the concentration of immunoglobulins did not differ between first and second time point. This is in contrast to other studies and is probably related to the great diversity of study population and may be seen as a bias for the results.
Protein concentration
Authors should explain why they performed these analyses because the lower content of protein in DBM is already known.
Line 224 Table S2 as mentioned above the number of infants per gestational age seems too small for any comment on gestational age effect.
Figure 2 authors conclude that gastric digestion reduces the amount of immunoglobulins and more so in DBM compared to MBM. However looking at figure 2 B, C and D it seems that the amount of SIgA, IgM and IgG of the group DBM is increased after gastric digestion. How do authors explain this finding?
Discussion
The text is mainly a repetition of the findings and could be more to the point concerning wat this study adds.
Author Response
Reviewer 2
Demers-Mathieu et al present a study that aimed to evaluate differences in immunoglobulin concentrations in human milk, gastric residuals and stool samples of preterm infants in relation to feeds of either mother’s own milk (MBM) or donor breast milk (DBM). The research objective is important since an impaired immune function is an important risk factor in prematurity.
Previously the same group presented comparable research, evaluating the effect of gastric digestion on either enzymatic activity, proteins and immunoglobulins in preterm and term infants. The current study adds the evaluation of donor breast milk and stool samples. Authors previously concluded that human milk total SC (SIgA/SC/SIgM), IgM, and Ig G, except IgA survived mostly intact after gastric digestion, especially in preterm infants. The current study found a lower amount of immunoglobulins in DBM compared to MBM, which was also seen in gastric samples. Somewhat in contrast to previous findings IgA was found most abundant in stools.
While the paper is well structured and easy to read, a number of major comments can be made with regard to theoretical assumptions and design of the study.
With regard to design: 20 and 16 paired samples from preterm infants with a gestational age ranging from 26 to 36 weeks were evaluated at 2 time points. This is a very small number taking into account the great maturational differences between very preterm infants and nearly term infants. Thus results may be biased by great interindividual differences.
>> We added a supplemental Figure (Figure S2) to show the immunoglobulin concentrations in milk and gastric contents from 3 gestational age (GA) groups of preterm infants (26–27 weeks of GA, 30–31 weeks of GA, 35–36 weeks of GA) fed mother’s own breast milk (MBM) and donor breast milk (DBM). We did not identify any particularly relevant differences between the GA groups that differed from the findings with the infants grouped across GA, which is likely due to the small number of subjects in each GA group (GA 26-27 n = 8, GA 30-31 n = 8 and GA 35-36 n = 4). We added this as a limitation of the study in the Discussion (lines 421-423).
A power calculation is not presented.
>>We added the power calculation used for the study in the method section (line 135-138).
Further, it is well- known that bio-active factors in human milk differ between mothers of very-/pre- and term infants and the content of these factors changes during the lactation period. A lower content of protein and immune- factors in DBM has also been demonstrated many years ago and is one of well-known disadvantages of DBM. Authors themselves previously demonstrated that gastric digestion will affect the feed content, thus in this respect the study does not present new insights.
>> One previous study (Ewaschuk et al., 2011) demonstrated that immunoactive components (IFN-γ, TNF-α, IL-1β, IL-10 and HGF) were significantly reduced by pasteurization from donated pooled milk from 34 mothers. However, the degree to which MBM and DBM antibody concentrations differ and how this affects survival of antibodies during gastric digestion were not previously demonstrated, therefore we examined these questions (lines 97-99).
Immunoglobulins in stool: while analysis of feed and gastric residuals are based on a single feeding event at two time points the stools are collected within 48 hrs after the feeding event. How do authors guarantee that the immunoglobulins measured in stool are related to the single feeding? How do authors know that the immunoglobulins found in stool originate from MBM of DBM and not form the infant self?
>> We did not assume that antibody measured in stool was from a single feeding.
We indicated that stool samples were collected within 48 h after the first feed and that its contents derive from a mixture of their different feeds. We added this statement to the Methods section (lines 151-153). We also added a sentence in the discussion to point out that antibodies detected in stool samples could derive from the MBM and/or DMB feeding or other previous feedings (line 389-390). We also added that a paragraph about the limitations of this study, in which we specified that we could not collect based on a specific feed in this study which is why we used a large 48-h convenience sampling window, and that future studies may be able to separate stools deriving from specific feeds using indigestible markers, such as food colorants (lines 413-418).
While it is known that the amount of immunoglobulins in preterm infants usually is lower than in term borns or adults it has also been demonstrated that human milk contains a number of bio-active factors that probably stimulate the production of immunoglobulins and thereby may increase the amount of own infant immunoglobulins. This seems in contrast to author’s assumption in the introduction that intestinal mucosal plasma cells only begin producing SIgA after 1 mo of age. Further, this assumption is not confirmed by reference [3]. Probably, to demonstrate the survival of immune-factors into stool radioactive labelling would be needed.
>>Thank you for pointing out our reference error. We have updated this section. Indeed, IgA-, IgM- and IgG-positive plasma cells were detected in the lamina propria of rectal and colonic biopsies of infants within the first month of age, thus it is possible that some of the immunoglobulins detected in the infant stool could derive from the infants own production. Thus, a protein labelling approach (radioactive or stable isotopes) would be needed to confirm that Ig in stool derive from milk. We have added this limitation to the discussion section (lines 418-423).
Materials and Method
- range of prematurity: did author consider to evaluate sub groups base on Ga for instance 26-28, 29- 30, 30 – 32. However, this would probably have meant to include a higher number of patients.
>> As mentioned above, we added a supplemental figure (Figure S2) showing a comparison of subgroups based on GA (26–27 wk, 30–31 wk and 35–36 wk) fed MBM and DBM and referenced this in the Methods section. We did not identify any particularly relevant differences between the GA groups that differed from the findings with the infants grouped across GA, which is likely due to the small number of subjects in each GA group (GA 26-27 n = 8, GA 30-31 n = 8 and GA 35-36 n = 4). We added this as a limitation of the study in the Discussion (lines 421-423).
- eligibility: how many infants were suspected for EOS and received antibiotics postnatally? This may have changed the intestinal microbiome and the amount of intestinal bio-active factors.
>>5 infants out of 20 were given antibiotics postnatally. We added this information to Table 1. We evaluated using Student’st-test the effect of antibiotics on the antibody concentration in gastric contents and stools samples from preterm infants. We did not find significant differences between infants that received antibiotics and those without antibiotics for SIgA, total IgA, IgM and IgG in gastric contents and stools samples from preterm infants after MBM or DBM feeding. We found that infants without antibiotics tended (p = 0.086) to have a higher SIgA concentration (2.5-fold) in gastric contents after MBM feeding than those that received antibiotics. Antibiotics may have affected the SIgA concentration in gastric digestion. We also added this information in the discussion (lines 402-407) and added supplemental Table S2 showing the p-values of Student’s t-tests comparing infants with and without antibiotics.
Line 99 with fortifier on ‘d’? probably day
>> We changed all instances of “d” to “days“.
- how many ml feeding did infants receive?
>> We added the volume of feeding (16 ± 12 mL for 8-9 days of postnatal age and 28 ± 10 mL for 21-22 days of postnatal age) in the Table 1.
Stool collection time: see above, motivation for this procedure is needed. Was the first stool after the event collected?
>> See explanation above.
Statistical analyses
- as mentioned above it is very strange that the concentration of immunoglobulins did not differ between first and second time point. This is in contrast to other studies and is probably related to the great diversity of study population and may be seen as a bias for the results.
>>We added a supplemental figure (Figure S1) to show the concentration of antibody between first and second time point. We observed a slight decrease (but not significant) from the first week to the second weeks in human milk antibodies. As these p-values are close to significant, it is likely that by increasing the number of infants in our study we would be able to observe these differences. We have mentioned this as a study limitation on lines (434-436).
Protein concentration
Authors should explain why they performed these analyses because the lower content of protein in DBM is already known.
>> We included the protein content just as a nice baseline measurement even though this is already known. Our findings also serve as a baseline validation that protein concentration is lower in donor milk than in preterm milk is a validation of the previous study (John et al., 2019). We believe that provide results that validate previous studies is worthwhile. We added this comment (lines 334-335).
Line 224 Table S2 as mentioned above the number of infants per gestational age seems too small for any comment on gestational age effect.
>> We removed the linear regression results (Table S2) on gestational age effect from this study.
Figure 2 authors conclude that gastric digestion reduces the amount of immunoglobulins and more so in DBM compared to MBM. However looking at figure 2 B, C and D it seems that the amount of SIgA, IgM and IgG of the group DBM is increased after gastric digestion. How do authors explain this finding?
>> This slight, but not significant, increase observed could be due to the remaining MBM feeding (residual feeding) in the gastric contents from preterm infants. We added this comment in the discussion (lines 375-379).
Discussion
The text is mainly a repetition of the findings and could be more to the point concerning wat this study adds.
>>For the first time, this investigation reported that total IgA, SIgA, total IgM and IgG concentrations were higher in the stomach from preterm infants fed MBM than those fed DBM. This higher concentration could be due to the initial higher concentration of maternal antibodies in MBM compared with DBM. We also found that the proportions of Igs from MBM in infant stool differed significantly from those in feed and gastric contents, especially the proportion of total IgA made up by SIgA. Unlike MBM and DBM samples in which most of the total IgA was SIgA, most of the total IgA in stool was IgA without the SC. IgM and IgG were detected in the stools at concentrations lower than total IgA.
These findings indicate that the concentration of maternal antibodies in DBM may be lower than in MBM. If that is the case,preterm infants fed DBM may receive lower amounts of antibodies than infants fed MBM. Lower ingestion of milk antibodies could result in lower protection against infections in these vulnerable infants. We believe these points have been clearly articulated in the paper.